# Delivery Strategies of Probiotics from Nano- and Microparticles: Trends in the Treatment of Inflammatory Bowel Disease—An Overview

**DOI:** 10.3390/pharmaceutics15112600

**Published:** 2023-11-08

**Authors:** Sílvio André Lopes, Cesar Augusto Roque-Borda, Jonatas Lobato Duarte, Leonardo Delello Di Filippo, Vinícius Martinho Borges Cardoso, Fernando Rogério Pavan, Marlus Chorilli, Andréia Bagliotti Meneguin

**Affiliations:** 1Department of Drugs and Medicines, School of Pharmaceutical Sciences, Sao Paulo State University (UNESP), Araraquara 14800-903, Brazil; silvio.lopes@unesp.br (S.A.L.); jl.duarte@unesp.br (J.L.D.); leonardo.filippo@unesp.br (L.D.D.F.); vinicius.mb.cardoso@unesp.br (V.M.B.C.); fernando.pavan@unesp.br (F.R.P.); marlus.chorilli@unesp.br (M.C.); 2Vicerrectorado de Investigación, Universidad Católica de Santa María, Arequipa 04000, Peru

**Keywords:** inflammatory bowel disease, gastrointestinal tract, probiotics, drug delivery systems, microparticles, nanoparticles

## Abstract

Inflammatory bowel disease (IBD) is a chronic inflammatory disorder, most known as ulcerative colitis (UC) and Crohn’s disease (CD), that affects the gastrointestinal tract (GIT), causing considerable symptoms to millions of people around the world. Conventional therapeutic strategies have limitations and side effects, prompting the exploration of innovative approaches. Probiotics, known for their potential to restore gut homeostasis, have emerged as promising candidates for IBD management. Probiotics have been shown to minimize disease symptoms, particularly in patients affected by UC, opening important opportunities to better treat this disease. However, they exhibit limitations in terms of stability and targeted delivery. As several studies demonstrate, the encapsulation of the probiotics, as well as the synthetic drug, into micro- and nanoparticles of organic materials offers great potential to solve this problem. They resist the harsh conditions of the upper GIT portions and, thus, protect the probiotic and drug inside, allowing for the delivery of adequate amounts directly into the colon. An overview of UC and CD, the benefits of the use of probiotics, and the potential of micro- and nanoencapsulation technologies to improve IBD treatment are presented. This review sheds light on the remarkable potential of nano- and microparticles loaded with probiotics as a novel and efficient strategy for managing IBD. Nonetheless, further investigations and clinical trials are warranted to validate their long-term safety and efficacy, paving the way for a new era in IBD therapeutics.

## 1. Introduction

Inflammatory bowel disease (IBD) is a chronic inflammatory disorder of the gastrointestinal tract (GIT) characterized by unusual immune responses associated with the microbiota of the digestive system in genetically susceptible patients [1,2]. With an estimated 3.1 million individuals affected in the United States alone and an increasing global incidence, IBD presents a substantial public health challenge. Ulcerative colitis (UC) and Crohn’s disease (CD) are the two main forms of IBD, which affect various regions of the intestine, particularly the ileum and the large intestine [3,4]. Environmental, genetic, and behavioral factors (lifestyle) may also be related to the onset of these diseases [4]. The course of IBD can be marked by alternating periods of occurrence and remission of the symptoms [4].

The conventional therapeutic options for IBD involve oral administrations of anti-inflammatory drugs, with mesalazine as the first choice, which aim to maintain remission time as long as possible and prevent episodes of inflammation. If left untreated, IBD can exacerbate with more intense symptoms such as rectal bleeding, anemia, gastrointestinal spasm, nausea, fever, fatigue, and loss of weight and appetite [5]. In more advanced cases, IBD patients may show epithelial erosions, ulcerations, and fibrosis [6] that may progress to colorectal cancer [7].

In the current therapy for IBD approved by the US Food and Drug Administration, mesalazine is incorporated into dosage forms based on specific polymers that present several release mechanisms. For example, it includes Asacol (pH-responsive), Colazal (a prodrug with enzymatic reduction), Mezavant (both pH-responsive and prolonged-release), and Pentasa (prolonged-release) (6, 18). The effectiveness of these systems is limited [8]; as many as two-thirds of patients do not respond satisfactorily and require surgery [9]. Also, in this therapy, the anti-inflammatory is administered in high doses and for prolonged periods of time, which leads to systemic adverse side effects [9]. The low efficacy of the treatment and the severe adverse effects occur due to a rapid absorption of most parts of the drug in the jejunum portion, with the remaining portion being released continuously throughout the GIT, before the inflamed site of the intestine. Therefore, only about 20% of the administered drug is effectively released into the colon [10].

This review article aims to provide an in-depth exploration of the potential of nano- and microparticles loaded with probiotics as a cutting-edge approach to IBD treatment. Several studies have shown the important benefits of using probiotics as adjuvants to the traditional treatment of IBD [11]. The use of probiotics was associated with symptom remission or attenuation, angiogenesis blockage, inflammation reduction, intestinal barrier reinforcement, and a reduction in the time to recover from the disease [12,13,14,15,16,17,18,19]. However, their therapeutic potential has been hindered by challenges such as limited stability, viability, and targeted delivery. The strategy of drug delivery directly into the colon, through micro- and nanoparticles, has been studied in the treatment of IBD [20]. The use of these structures is potentially promising due to the ability of these particles to penetrate the mucosal layer and the intestinal epithelium affected by the disease [21]. These particles protect the drug from degradation and allow it to reach the affected colon in high concentrations, conferring prolonged and maximized pharmacological effects and minimizing the adverse side effects [8]. Thus, there is a growing demand for the development of new systems based on micro- and nanotechnologies [9]. Not only anti-inflammatory drugs but also probiotics could benefit from these new delivery systems.

In sum, this review seeks to shed light on the innovative potential of probiotic-loaded particles as a promising avenue for the management of IBD, providing a comprehensive overview of the current research and future prospects in the field.

## 2. Inflammatory Bowel Disease (IBD)

The main diseases classified as IBD are ulcerative colitis (UC) and Crohn’s disease (CD) [1,2], both characterized by the chronic relapse and remission of inflammation in the GIT [22]. These two subtypes differ in the extension of affected regions and the most common symptoms.

### 2.1. Ulcerative Colitis

Ulcerative colitis (UC) was recognized as a disease for the first time in 1859 by Samuel Wilks [23]. It is characterized by diffuse inflammation of the large intestine, restricted to the mucosa and submucosa and limited to the colon and rectum. The mucosal involvement is continuous and ascending, and the transition between affected and normal tissues is well demarcated [24]. The classification of UC varies according to the symptomatology, as shown in Figure 1. Depending on the extent of the affected colon, colitis is classified as proctitis or left colitis, with both conditions affecting parts of the colon, or pancolitis, affecting the entire colon. Proctitis affects around 30 to 60% of patients with UC and left colitis is observed in range between 16 and 45% of the affected people. Proctitis is the situation in which colitis affects the rectum, associated or not with rectal bleeding and tenesmus. Left-side colitis can affect the rectum and the sigmoid and descending colon, and is associated with bloody diarrhea and severe abdominal cramps. Finally, extensive colitis (or pancolitis) is the condition in which colitis affects the whole organ and is present in 15 to 35% of patients. It includes all the above characteristics plus fatigue and fever [25].

The main feature of extensive colitis is diarrhea with the rapid flow of intestinal contents through the inflamed colon causing ulcerative lesions on the inner wall of this organ. Diarrhea crises can be postprandial or even nocturnal, accompanied by rectal bleeding, abdominal pain, fever, weight loss, fatigue, tenesmus (incomplete sensation of evacuation), and general malaise; these are symptoms that may progress to an even more serious clinical picture [24]. The greater the severity of the inflammation, the greater the amount of diarrhea, a mixture of blood and feces. More severe cases of UC, involving severe bleeding, occur in about 10% of patients with urgent colectomies, and about 1% of all affected people have at least one occurrence of massive bleeding, which may require surgical intervention [26]. The main complications of UC comprise symptoms such as toxic megacolon, enterorrhagia, and severe acute colitis [24]. Higher risks of developing colorectal cancer are also associated with UC [7], which may increase by 2.2% in general cases of IBD and 7.0% in cases of long-term UC [27].

### 2.2. Crohn’s Disease 

Crohn’s disease was first reported in 1932 by Burrill B. Crohn, Leon Ginzburg, and Gordon D. Oppenheimer [28]. CD can discontinuously reach any region of the GI tract, from the mouth to the anus, most frequently the ileum and colon [3]. The most common symptoms are recurrent bouts of diarrhea, fever, severe abdominal pain, and weight loss, impairing the life quality of the patients.

At the beginning of CD, there is hypertrophy of the intestinal mucosa and submucosa, which modifies the normal transverse folds, changing the appearance of hemorrhagic ulcers, which evolve into fissures. In the chronic phase, there are edemas in the submucosa and fissured ulcers. Thus, CD causes histological alterations, the formation of edemas, a reduction in mucus-producing cells, hyperplasia of intestinal crypts, erosions, deep ulcerations, and lymphoid aggregates. In cases of complications, intestinal granulomas, fistulas, stenoses, abscesses [27], and intestinal perforation may occur, a pattern that is not seen in UC. In CD, an increased activity of T helper cells 1 (Th1) and 7 (Th17) leads to dysfunctional inflammation and alters the response of the intestinal immune system, resulting in the excessive release of cytokines such as TNF-alpha, interferon-gamma, IL-12, IL-3, and IL-17 [29]. Therefore, an environmental stimulus in a person who has a genetic predisposition to CD can induce an immune imbalance of Th1 or 17 in relation to regulatory T cells (Treg cells), a situation that may favor the onset of the disease [30].

### 2.3. Immunopathogenesis

Impairments in the integrity of the intestinal epithelial barrier may be the first step towards the onset of IBD. In immunologically healthy individuals, communication between the epithelial cells of the innate and adaptive immune system ensures intestinal homeostasis [31]. The epithelium, composed of the epidermis and the intestinal mucosa, is the first barrier to defend the body against the invasion of pathogens [32]. Through signaling processes, the epithelium remains in contact with the intestinal microbiota in a state of immunological tolerance, which is maintained by constant sampling through antigen-presenting cells (APCs) [33] of luminal material from the intestine, such as antigens, which triggers the immune response, regulated by the T lymphocytes (T reg cells) [27]. Studies in animal models indicate that the balance between Th17 and Treg cells, which are responsible for the production of pro-inflammatory and anti-inflammatory cytokines, is essential for intestinal homeostasis and is directly affected by the content of the normal microbiota [34].

In a healthy individual, the epithelial layer is renewed every two to three days. Renewal includes apoptosis and enterocyte desquamation of the crypt regions of the colon and the villi structures of the small intestine. If this renewal process is interrupted, the epithelial barrier can be impaired and chronic inflammation can occur [35]. Genetic, immunological, but also microbiological variations affect renewal balance, with disturbances in this process contributing to the onset of gastrointestinal disorders, such as IBD [31]. Thus, defects in the colonic epithelial layer may promote the pathogenesis of IBD through the free passage of microorganisms and triggering the onset of an immune response [35].

In addition to the enterocytes, the microbiota signals the innate immune cells in the lamina propria through the pattern recognition molecule signal receptors [35]. The lamina propria can be understood as a constituent region of the mucosa, present just below the intestinal epithelium, which is reinforced by loose connective tissue and occupied by resident immune cells, such as macrophages and dendritic cells [36]. To enable this mechanism, the recognition of microbial antigens through the action of Toll-like receptors (TLRs) is fundamental. In addition to these receptors, the presence of NOD-like cytoplasmic receptors is essential to initiate the phagocytosis of these microorganisms [37]. Failures in this mechanism result in deficiencies in eliminating bacteria and their products, leading to pathogenic microbial invasions and the progression of IBD [38]. These signals have been shown to be necessary for normal homeostasis and resistance to injury. Cytokines, such as interferon-gamma (IFN-γ), interleukins IL-17 and IL-22, and pathogenic organisms, disturb the epithelium by adjusting the function of the barrier [35]. Thus, UC evolution is associated with changes in the immune homeostasis of the intestine resulting from defects in the mucosal barrier in the affected region, which increases tissue permeability, leading to endotoxin secretion and bacterial translocation, contributing to the establishment of chronic inflammation [31,32].

The colonic epithelial barriers can also be damaged by a decrease in epithelial resistance and of paracellular junction proteins such as claudins and occludins. Consequently, the permeability to ions and water at the injured sites increases. The appearance of lesions, apoptosis, and mucosal necrosis is related to an increase in the release of reactive oxygen species (ROS), tumor necrosis factor (TNF-α), IL-1B, and IFN-ɣ. The process of apoptosis can allow erosions, ulcers, local leaks, increased transcytosis, and the increased uptake of antigens and bacteria [6]. In the exacerbation phase of IBD, there is an increase in the production of pro-inflammatory cytokines such as IL-1, IL-6, IL-8, and TNF-α [39]. In addition, disturbances in intestinal epithelial cells, such as Paneth cells [40], which produce antimicrobial substances and growth factors [35], are related to important genetic risk factors for IBD and can lead to intestinal inflammation [40].

There are many factors explaining the worsening of IBD. The TLR4 signaling pathway, a subtype of Toll-like receptors, influences this process. Lu et al., reported that TLR4 causes destructive effects on intestinal epithelial tissues and ulceration in patients with UC due to the release of pro-inflammatory cytokines which aggravate intestinal inflammation [41]. Through the activation of TLR4, the inflammation process can favor the development of pathogenic bacteria in addition to changes in the composition of the colonic microbiota. Therefore, the immune system can induce an intense reaction, stimulating the expression of cytokines and favoring the progression of the disease [42].

Because IBD presents disordered immunological responses, a striking characteristic is the chronicity of inflammatory processes. Therefore, immunopathogenesis is one of the most important therapeutic targets in IBD. The identification of new genes reinforces the central roles of innate and adaptive immune cells, as well as autophagic processes in the pathogenesis of IBD. Therefore, autophagy is a crucial point in its pathogenesis, demonstrating the link between changes in immunity and pathogenic factors [43]. In addition, autophagy modulates the balance of pro-inflammatory and anti-inflammatory cytokines through different pathways, different cells in the innate immune system, and non-immune cells, so many immunological changes are linked to this process. Defective autophagy can incite inflammation in the gut region, altering cytokines and innate and adaptive immune cells in IBD. It is believed that effective autophagy can ensure the balance between intestinal tolerance and immune tolerance homeostasis. Thus, autophagy is a potential target for the treatment of IBD, especially when current treatment methods are mainly immune-related therapies [43].

Although the lymphatic system is involved in the initiation and progression of IBD, more studies are needed to better understand this mechanism of action in relation to IBD. In this sense, studies with animal models have investigated the lymphatic system in IBD, showing ascending evidence of the involvement of lymphoid tissues in the pathogenesis of IBD. Lymphadenopathy, a condition described by an increase in the number and size of mesenteric lymph nodes (MLNs), is commonly seen in patients with IBD. The appearance of IBD, stimulated by lymphocyte–dendritic cell interaction, is related to the presence of B and T lymphocytes between the gut-associated lymphoid tissue (GALT) and the intestinal mucosa, induced by lymphocyte–dendritic cell interaction. Furthermore, the luminal microbiota also colonizes lymphoid tissues and can stimulate new local immune responses [44].

### 2.4. Predisposing Factors

Several environmental and physiological factors, patients’ habits, and psychological factors are associated with the occurrence of IBD, as described in Figure 2.

Studies suggest that Caucasian people and inhabitants of urban and industrialized regions are more likely to be affected by autoimmune diseases in general, including IBD [45]. In the highly industrialized countries of North America and Western Europe, IBD is considered a serious public health problem [46]. From 2012 to 2017, the annual incidence ranged from 15.7 cases per 100,000 people in North America to 24.3 in Europe [1,47]. It has been observed that even in areas traditionally of low prevalence, like Brazil, South Korea, and China [34], there has been an increase in the incidence of IBD after processes of urbanization and development. In Brazil, there has been a marked increase in the incidence of CD over the last 40 years. The incidence, which was 0.08 in 1988, increased to 5.48 in 2015 per 100,000 people [48]. Newly industrialized countries in Asia, Africa, South America, and the Middle East have also documented a progressive increase in IBD in recent decades [25,49], at rates higher than 14.9% [36], far above the rates of Europe and North America, which is further evidence that the occurrence of IBD is associated with industrialization and urbanization activities [1,50], and probably as a result of the exposition of a usually increased rate of air pollution [38].

A deficit in vitamin D, important in calcium metabolism, may favor the onset of IBD [51], and sudden changes in diet; the use of medications such as oral contraceptives; hormone replacement therapy; nonsteroidal anti-inflammatory drugs; physical and mental stress [25]; tobacco consumption [38,51]; and drugs such as Aspirin, Clozapine, Entocapone, Lansoprazole, Omeprazole, Ranitidine, and Sertraline, among others, are cited as predisposing and risk factors. Physical activity, opposite to sedentarism, was associated with a decreased risk of CD occurrence by 44%, and sleep disorders could be a determinant in the development of the active phase of the disease [51]. In addition, between 5 and 15% of adult IBD patients can develop chronic kidney disease over time [52].

As factors that can trigger IBD, alcoholism has a significant effect on in-hospital mortality in IBD patients. Although uncommon in people with IBD, abstaining from alcohol use is highly suggested given the significant risk of mortality associated with alcohol abuse. Furthermore, like IBD, it causes chronic pain and consequent damage to quality of life, which can be a predisposing factor to alcohol abuse [52].

Historically, the 1930s was the time that gastroenterologists and psychiatrists first suggested that emotional life events and experiences could be related to the exacerbation of intestinal symptoms [53], as well as in relapses [54]. Stress processes can stimulate alterations to the motor, secretory, and sensorial functions of the GIT, as well as the thresholds for pain perception, favoring intestinal permeability and inflammatory exacerbation. It is believed that in almost 75% of patients with IBD, events of stress or the own personality of the patients are main contributors to symptom development [55].

It is understood that the consequences of psychological stress to intestinal health are mediated by neuroendocrine-immune pathways related to the brain–gut axis, specifically the hypothalamic–pituitary–adrenal axis (HPA) and the sympathetic nervous system, which can interact directly with the immune system [55], leading to the production of pro-inflammatory cytokines, the activation of macrophages, and changes in permeability and intestinal microbiota [38,51]. Not only are the profiles of cytokines IL-1β, IL6, IL10, IL4, and TNFα altered by stress, but that of the hormone cortisol is too [56,57]. The mucosal mast cells can be activated by stress, thanks to their communication with neurons in the GI tract, through the release of eicosanoids, serotonin, and IL6, situations that may favor the onset of IBD [55].

Psychological stress stimulates corticotropin-releasing factor (CRF) secretion from both central and peripheral parts of the CNS (hypothalamus and adrenal cortex, respectively). Central CRF regulates the ACTH-cortisol system, while peripheral CRF directly influences gastrointestinal motility. In general, CRF, through stress as a trigger, inhibits upper GI motility and stimulates colon motility, so that symptoms such as abdominal pain and changes in bowel function can occur in IBD without significant disease activity [55]. In addition, IBD and stress may be associated with structural and functional changes in limbic structures that may result in the exacerbation of symptoms due to regulation of the autonomic and endocrine pathways of the brain–gut axis [58,59]. Psychological stress can also increase intestinal permeability, possibly as a result of changes in the cholinergic nervous system and mucosal mast cell function. This altered permeability reduces mucosal barrier function and alters host–bacterial interactions [55].

## 3. Conventional Treatment of IBD

5-aminosalicylic acid, also known as 5-ASA or mesalazine and belonging to the aminosalicylates class, is the drug of first choice for the treatment of mild to moderate IBD [60]. The structural formula and mechanism of action of 5-ASA is shown in Figure 3.

This drug is administered orally, through enteric-coated capsules or tablets; parenterally, through applications of subcutaneous or intravenous injections; or rectally, through suppositories or enemas [6]. It is believed that 5-ASA acts on the intestinal mucosa through a combination of anti-inflammatory effects, involving the inhibition of IL-1 production [61], blocking the production of IL-2 and the inhibition of T cell proliferation and [62], the reduction of TNF-α receptor signaling [63], the inhibition of cyclooxygenase (blocking the production of prostaglandins) and 5-lipoxygenase (blocking the production of leukotrienes) [64,65], and the activation of adhesion molecules in the endothelial cells [66]. It also possesses antioxidant actions that eliminate free radicals [67].

IBD requires long drug therapy, in many cases for indefinite periods. For therapeutic success, total patient compliance with treatment time and correct use of the prescribed drugs are essential, but this is an uncommon situation [45]. Many studies worldwide show a low adherence (40–60%) of patients affected by IBD to 5-ASA treatments [68,69,70,71,72]. Reasons are forgetfulness, bad feeling, the frequency of medication dosage, lack of medication effect, medication unavailability, side or adverse effects, lifelong treatment, and treatment cost [73].

There are situations in which patients do not respond to the use of aminosalicylates [74]. In moderate or severe cases, the administration of corticosteroids, such as dexamethasone, thiopurines (azathioprine), or calcineurin inhibitors (cyclosporine) is common [6,75]. Prednisone [74,76] and budesonide also are employed [74,77,78]. Weight gain, edema, insomnia, emotional lability, psychosis, acne, osteoporosis, cataracts, glaucoma, stretch marks, myopathy, susceptibility to infections, hepatic steatosis, systemic arterial hypertension, *Diabetes mellitus*, and acute pancreatitis are reported as side effects [79]. There are patients resistant to steroids, requiring the administration of immunosuppressive drugs such as azathioprine, tacrolimus, or methotrexate [74,76]. The use of azathioprine may, in turn, cause fever, nausea, vomiting, abdominal pain, medullary depression, changes in liver enzymes and an increased risk of non-Hodgkin’s lymphoma, and of methotrexate alopecia, pneumonitis, changes in liver function, liver fibrosis, and teratogenicity [79].

In patients with marked UC activity, parenteral treatment consists of the use, at the hospital-level, of corticosteroids, biological therapy, and cyclosporine. Depending on the situation, infliximab, an anti-TNFα, may be administered [79]. Treatment with azathioprine or anti-TNF-α immunotherapy or vedolizumab is used in cases where the disease is frequently relapsing [27]. If patients do not respond to drug therapy and the disease evolves to extreme cases of intestine perforation and hemorrhagic and toxic megacolon, a surgical intervention (colectomy) may be required [79].

The oral route is the most accepted for drug administration and is the way used for approximately 60% of small molecule drugs that are commercially available [80]. It is noninvasive, safe, and cost-effective, factors that contribute to increased patient adherence [5,80]. In the specific case of 5-ASA, the drug is rapidly and extensively absorbed in the upper portions of the GIT, which results, as already mentioned, in several side effects, in addition to reduced concentrations in the colon (the main affected region), decreasing the effectiveness of IBD treatment. In view of this, the incorporation of the drug to specialized carriers that allow its targeting of the inflamed regions is necessary. This is possible only with the development of drug delivery systems that are specific to the colon. In addition to a significant reduction in unwanted distribution of the drug into the body, such a system would allow for dose reduction, a decrease in adverse and toxic effects, and maximum concentration of the drug in the desired target [5].

The main treatment of mild to moderate cases of IBD involves the use of orally administered drugs. However, the effectiveness of conventional drugs is hampered by several factors, including severe adverse effects, lack of targeting of the inflamed tissue, and others, that are associated with the anatomical and physiological structure of the GIT. These difficulties make the knowledge of the distinct portions of the GIT of paramount importance during the search for alternatives to better treat IBD and for effective targets for drug delivery into the colon.

New and emerging treatments for IBD are being studied as new therapeutic approaches, such as the Inhibition of Immune Cell Trafficking, Sphingosine-1-Phosphate Modulators, the Inhibition of Cytokines, JAK Inhibitors, TLI1A Inhibitors, Phosphodiesterase Inhibitors, and IL-36 Inhibitors [81]. In recent years, many new medications have been developed and therapy for IBD patients has changed significantly. However, current therapies still have unresolved problems, which makes their adherence difficult and contributes to the increase in new numbers of IBD cases around the world. Therefore, new therapies are tremendously urgent [81].

## 4. Gut Microbiota, Dysbiosis, and the Potential Use of Probiotics in IBD Therapy

The colon has been investigated as a target for the release of drugs [82] and probiotics for local and systemic action. However, the colon as a target poses a great challenge since it is the most distant portion of the GIT. Drug delivery systems with high specificity for the colon are, therefore, necessary [83]. Since this region is a place of great enzymatic activity resulting from the extensive colonic microbiota, it has been successfully explored as a trigger to biologically degrade specific polymers, promoting drug delivery [83].

### 4.1. The Gut Microbiota

The intestinal microbiota, also known as the gut microbiota or gut microbioma (GM), is a complex and diverse community of microorganisms that inhabit the human digestive tract [82]. This community includes bacteria (Eubacteria and Archaea), viruses, and fungi [84]. In healthy individuals, the GM plays important roles in maintaining the health of the digestive system and the body as a whole. In addition to aiding in digestion, the GM is also involved in a variety of other physiological processes, including immune system regulation and activation, metabolism, and even brain function [59,84]. GM disturbances are involved in several neurological disorders such as depression, anxiety, multiple sclerosis, schizophrenia, [59] anorexia nervosa, Alzheimer’s disease, and Parkinson’s disease [85].

GM metabolites also contribute to beneficial interactions within the gut microorganisms themselves [86]. The healthy GM works as a homeostatic organ participating in the synthesis of short-chain fatty acids (SCFAs), in processes of fermentation of undigested polysaccharides, in the synthesis of vitamins (such as B and K), in the production of energy, and in the prevention of infections caused by pathogenic microorganisms [34,84]. The SCAFs, produced in the colon, are composed of propionate, butyrate, and acetates. Their synthesis occurs through the fermentation of undigested carbohydrates by obligate anaerobes (mainly Firmicutes and Bacteroidetes), contributing to bacteria that ferment oligosaccharides (e.g., Bifidobacterium) [86]. Butyrate is a primary source of energy for colonic epithelial cells, so, when added to propionate, both can regulate intestinal physiology and immune function. In turn, acetate acts as a substrate for lipogenesis and gluconeogenesis [86]. The predominance of obligate anaerobic bacteria in the colon, belonging to the phyla Firmicutes and Bacteroidetes, provides a strict anaerobic environment. Although colonocytes are the main cellular source of oxygen in the colon, the diffusion of oxygen from colonocytes to the lumen is strictly limited. While superficial colonocytes contain less than 1% oxygen, host tissues contain between 3% and 10% oxygen. Intracellular hypoxia of colonocytes occurs due to the consumption of oxygen inside them through the mitochondrial β-oxidation of butyrate, originating from bacteria, and the production of carbon dioxide, which represents their main energy production pathway. The reduction in butyrate-producing bacteria decreases luminal butyrate levels, resulting in a metabolic reorientation of superficial colonocytes toward anaerobic glycolysis and increased oxygen diffusion into the lumen, thus leading to luminal aerobic and/or facultative anaerobic expansion through aerobic breathing [86].

Unlike the stomach and upper part of the small intestine, where few microorganisms are present, the number and diversity of bacterial species progressively increases from the jejunum to the colon [87]. The number of microorganisms that inhabit a healthy human gut can reach 10^14^ CFU/mL [88]. In the case of bacteria, there are more than one thousand species, predominating representatives of the phyla *Bacteroidetes*, *Firmicutes*, *Actinobacteria*, *Proteobacteria*, *Fusobacteria*, *Spirochetes*, *Tenericutes*, and *Verrucomicrobiota*, including the genera *Bacteroides*, *Peptococcus*, *Bifidobacterium*, *Clostridium*, *Eubacterium*, *Streptococcus*, *Enterobacterium*, *Enterococcus*, *Lactobacillus*, and *Fusobacteria* [34,84,89]. These bacteria release large amounts of enzymes responsible for deglucuronidation, decarboxylation, the reduction of double bonds, and hydrolysis reactions [83], all important to the body’s metabolism.

In recent work, the importance of bacterial tryptophan metabolism in the human intestinal context was demonstrated [86]. Tryptophan is an amino acid that plays crucial roles in the balance between intestinal immune tolerance and the maintenance of the intestinal microbiota. Tryptophan can be converted into indole-containing bioactive metabolites (indole, indolic acid, skatole, and tryptamine) by intestinal bacteria, and indole derivatives affect the host by triggering the aryl hydrocarbon receptor (AhR). AhR signaling enables host immune homeostasis through the secretion of interleukin (IL)-22 from CD4+ T cells and innate lymphoid cells in the intestine. IL-22 can trigger the release of antimicrobial peptides and modulate microbial composition. Furthermore, AhR activation plays an anti-inflammatory function by developing intraepithelial lymphocytes and innate lymphoid cells [86].

### 4.2. Dysbiosis

Dysbiosis is any change in the composition or function capacity of the intestinal microbiota, commonly associated with several intestinal pathologies [90]. In patients affected by IBD, when compared with non-affected individuals, there are reports that show that this imbalance comes from the reduction in the number of bacteria that have anti-inflammatory activity and the increase in the number of bacteria with inflammatory activity [91]. This unbalance results in function changes, damages in the fermentation process, and changes in biochemical processes leading to immunological disequilibrium and even drug release [6,34]. Diet, age, host genetics, exercise, antibiotic consumption, smoking, and geographical impacts are factors that can modify the composition of the gut microbiota [84].

The dysbiosis profile most commonly related to patients with IBD is a decrease in the Firmicutes phylum (*Faecalibacterium*, *Blautia*, *Roseburia*) [92], in commensal bacterial species in the genera *Firmicutes* (*Clostridium* and *Enterococcus*), *Bacteroides*, and bacteria from the genera *Lactobacillus* and *Eubacterium*. On the other hand, an increase is observed in phylum Proteobacteria belonging to the *Enterobacteriaceae* family (for example, bacteria such as *Escherichia* and *Shigella*) [34,92] and there is a considerable increase in bacteria from the phylum *Bacteroidetes* [93]. In addition, inflammation favors the growth of pathogenic bacteria like *Citrobacter rodentium* and *Salmonella* spp. [92].

Although the association of IBD with the presence of pathogenic species is not clear, adherent-invasive *Escherichia coli* (AIEC) was detected in a higher frequency in patients with Crohn’s disease (22%) than in healthy patients (6.2%) [94]. As mentioned, bacteria belonging to the genera *Clostridia* and *Bacteroides*, which are capable of causing anti-inflammatory and T-reg cells responses [95], and polysaccharide A from *Bacteroides fragilis* [96], were detected at higher incidences in the colons of IBD-affected people. Conversely, *Faecalibacterium prausnitzii*, which belongs to the *Firmicutes* phylum, was less frequent in patients with IBD than in healthy controls. This bacterium was associated with a lower severity of endoscopic recurrence after resection in humans and, in mice, minimized colitis symptom severity [97,98].

It has also been demonstrated that patients’ diet exerts a strong impact on the gut microbiota, which could influence disease incidence. Diets involving a high intake of animal proteins and saturated fat correlated with the presence of pathogenic *Bacteroides* spp., and those involving the consumption of carbohydrates and fiber correlated with the presence of beneficial *Prevotella* spp. [99,100]. In addition to bacteria, IBD may also involve imbalances in virus, Archaea, and fungi populations. Bacteriophages pertaining to the *Caudovirales* family were at higher incidence in patients with CD and UC, and the susceptibility of mice to chemically induced colitis was influenced by an altered immune response to indigenous fungi in the gut [101,102].

It is important to highlight that dysbiosis is intensified with the chronic use of medications designed to treat IBD, such as anti-inflammatory drugs, which includes the widely used drug 5-ASA [92,103], antibiotics, laxatives, and eating habits, which can alter the secretion of enzymes into the colon [6]. Through constant episodes of evacuation, IBD itself can also contribute to dysbiosis [104].

### 4.3. Potentials of Probiotics

One way to solve the dysbiosis problem is the use of probiotics. Probiotics can be understood as live microorganisms that, when administered in adequate amounts, confer health benefits [34,105]. The first comprehensive study that associated molecules produced and secreted by one microorganism with factors that could induce the growth of another microorganism was published in 1965 [106]. For a microorganism to be classified as a probiotic, it should be non-pathogenic to the host, be isolated from the same species, survive the transit through the GIT, and, finally, keep its viability during long periods of storage [107]. The most-used probiotics belong to the species of *Lactobacillus*, *Bifidobacterium*, *Escherichia*, *Enterococcus*, *Streptococcus*, *Saccharomyces*, and *Propionibacterium*. Most consist of lactic acid-producing bacteria such as lactobacilli, streptococci, enterococci, lactococci, and bifidobacteria. *Bacillus* spp., and fungi such as *Saccharomyces* spp. and *Aspergillus* spp. also have been used [107,108].

Probiotics provide important benefits by restoring or maintaining the balance of the normal intestinal microbiota, inhibiting the growth of pathogenic bacteria, improving the barrier function of the intestine, promoting local immunity, and interfering positively in intestinal inflammatory responses [105,109,110]. Regarding pathogenic bacteria, probiotics compete for adhesion sites through the production of bacteriocins and antimicrobial peptides, such as defensins, lysozymes, and phospholipases [109], by providing nutrients to the host body and by inhibiting the process of apoptosis in the intestinal epithelial cells. Overall, probiotics can improve the treatment of IBD and, consequently, the life quality of patients affected by these diseases [105]. Probiotics affect the immune system in several ways. They work by increasing the levels of cytokine and immunoglobulin, macrophage activation, increasing the activities of natural killer cells, autoimmune modulation, and immune stimulation against pathogenic bacteria and protozoa [14].

*Lactobacilli* was the first probiotic to be discovered. It belongs to the group of acid-lactic bacteria, comprising 183 known species, with various applications in industrial processes such as the manufacture of preservatives, food flavorings, medicines, and the manufacture of cosmetics. This genus is the dominant group in the gastrointestinal system. It produces large amounts of lactic acid and other metabolites using various substances as sources of carbon such as glucose, fructose, lactose, and galactose. They can be classified according to their metabolism as homofermentative and heterofermentative. The homofermentative group only produces lactic acid, and the heterofermentative group, in addition to lactic acid, produces several other metabolites, including ethanol, acetic acid, and carbon dioxide. *Lactobacilli* can also produce secondary metabolites, including bacteriocins, exopolysaccharides, and enzymes, which are used to increase the quality and shelf life of fermented foods [111].

The probiotics *Lactobacillus* spp. and *Bifidobacterium* spp. also produce the short-chain fatty acids (SCFA) acetate and propionate, which are organic acids that result from carbohydrate metabolism. These probiotics do not produce butyrate, obtaining it from commensal bacteria, such as *Faecalibacterium*, leading to increased levels of this compound in the intestine [110]. Butyrate promotes colon motility, reduces inflammation, increases visceral irrigation, inhibits the progression of tumorigenic cells, and induces the process of apoptosis [112]. SCFAs, in addition to acting as sources of energy, also reduce the pH in the intestinal lumen, hinder the development of pathogenic agents, influence intestinal motility, and reduce colon cancer by stimulating the apoptosis of tumorigenic cells [113]. SCFAs also act as signaling molecules by reducing the production of pro-inflammatory cytokines and increasing the population of regulatory T cells (Treg) in the large intestine [114].

Bifidobacteria are the most abundant microorganisms in the GIT. In addition to the benefits mentioned by other bacteria, *Bifidobacterium* spp. reduces intolerance to lactose, prevents GIT disorders, and decreases ammonia concentrations. The presence of an adequate environment and nutrients, known as “bifidogenic factors”, contributes to the viability and activity of bifidobacteria [107].

Direct benefits of probiotics in the treatment of IBD have been shown when several species of probiotics were combined (Table 1). The combination of probiotics *Lactobacillus* spp. and *Bifidobacterium* spp., used in commercial products [115], can be beneficial for the immune system, helping healthy and allergenic people by decreasing inflammatory reactions [14]. When *Bifidobacterium* plus *Lactobacillus* are combined with species of *Streptococcus* or with *E. coli* Nissle 1917, a remission of mild to moderately severe symptoms of UC was observed [15]. Another study reported a meta-analysis procedure to evaluate five clinical studies involving a total of 319 patients with UC in which the therapeutic response to a mixture composed of one strain of *Streptococcus thermophilus*, four of *Lactobacillus* spp., and three *Bifidobacterium* spp. strains, known as VSL#3, was compared with placebo. Symptoms of UC were attenuated in about 44.6% of subjects who were administered VSL#3, compared with a 25.1% reduction in the placebo group [16].

Probiotics have also been compared with the traditional use of 5-ASA. In a randomized clinical trial, *E. coli* strain Nissle 1917 was compared with 5-ASA to treat UC. The trial included 327 patients who were given either a daily dose of the probiotic or 500 mg of 5-ASA three times a day. After a period of twelve months, the patients were evaluated clinically, endoscopically, and histologically. *E. coli* strain Nissle 1917 was as effective as the administration of 5-ASA. A recurrence rate of 36.4% was verified in the *E. coli* Nissle 1917 group against a rate of 33.9% in the 5-ASA group [15]. Although the use of probiotics can be as effective as that of the traditional 5-ASA, the benefit of probiotics can be even greater when they are associated with commonly prescribed anti-inflammatories [116]. The association of 5-ASA with a combination of the probiotics *L. salivarius*, *L. acidophilus*, and *B. bifidum* BGN4 resulted in a reduced time of recovery of IBD symptoms and a more attenuated form of the disease. After more than 2 years of study, the results suggest that probiotic supplementation to conventional drugs should be performed without interruption and administered for a long time, allowing a viable treatment [18].

There is increasing evidence from experimental mouse models and from clinical observations that angiogenesis is an important component of the pathogenesis of IBD [117,118], and the yeast *Saccharomyces boulardii* was found to be associated with this process [13]. Chronically inflamed intestinal tissues exhibit changes in physiology and microvascular functions compared with intestine tissues not affected by IBD. Vascular endothelial growth factor (VEGF) and the VEGF receptor tyrosine kinase family (VEGFR) are proteins that modulate the process of angiogenesis. Transmembrane receptor VEGFR types 1 and 2 are present in endothelial cells and have a high affinity for VEGF. The binding of VEGF to these receptors leads to intracellular receptor phosphorylation which activates various intracellular signaling pathways, leading to endothelial cell proliferation and blood vessel formation [13]. VEGF was shown to be more concentrated in distal colon tissue in the colitis model [119], as well as in tissues and sera from patients with CD and UC [120,121,122]. Studies conducted in an experimental colitis model suggest that VEGF is an important mediator of IBD by promoting intestinal angiogenesis and inflammation [123]. Therefore, agents that inhibit VEGF/VEGFR signaling may be useful in reducing intestinal inflammation in patients with IBD, and *Saccharomyces boulardii* was found to block angiogenesis of VEGFR signaling and regulate the acute and chronic inflammation of the intestinal mucosa associated with the progression of IBD [13].

Several other reports on the benefits of probiotics to treat IBD exist. For instance, probiotics from the genera *Bacillus* are capable of surviving the hostile environment and low-pH in the upper GIT to reach the small intestine and secrete antimicrobial substances [17]. The yeast *Saccharomyces boulardii* and the bacteria *Akkermansia muciniphila* and *Faecalibacterium prausnitzii* induced reinforcement of the intestinal barrier and reduced colonic inflammation [19]. The population of the species *Akkermansia muciniphila*, although classified as a “next-generation” [115] microorganism, declined in IBD-affected individuals [89] and appeared to not resist storage and GIT conditions [115]. The bacteria *Enterococcus faecium* CRL 183 and *Bifidobacterium longum* ATCC 15707, when administered in the form of soy-based beverages, showed excellent results in the treatment of intestinal colitis. There was a lower degree of inflammation and ulceration in the colons of rats submitted to induced colitis in relation to the control groups, as well as an increase in the population of *Lactobacillus* spp. and *Bifidobacterium* spp. [12].

## 5. Probiotic Delivery Systems

To treat IBD, the delivery of probiotics directly into the affected sites would offer enormous benefits since it offers a restoration of the normal intestinal microbiota, attenuating the impacts of dysbiosis. However, delivering probiotics into the gut can be challenging, as they must maintain stability and viability, which can be affected by the harsh acidic environment of the stomach, and reach the intestines to be effective. To overcome this challenge, researchers have developed a variety of drug delivery ways such as micro- and nanostructured systems. Examples of works conducted to investigate drug and probiotic delivery systems are provided in the following sections.

### 5.1. Microparticle-Based Drug Delivery Systems

Microencapsulation covers protective conditions for bioactive compounds, cells, or bacteria. Although they are larger than nanoparticles, there is still no exact definition of the dimensions of each one, which can reach up to 1000 μm [124]. For its part, microencapsulation favors protection and release systems during its passage through the GIT [125]. There are several microencapsulation techniques, but in matters of scaling-up one of them would be spraying via atomization [126]. In food technology, ionic gelation has been widely used for temperature-sensitive compounds [127]. Several reported microparticles with potential GIT protective activity were reported including polymeric systems based on alginate, gelatin, chitosan, hyaluronic acid, pectin, cellulose, and fibrin, among others [128].

Recently, Zhu et al., 2022 [129] reported that during IBD a key target for the release of probiotics would be in specific regions of high production of reactive oxygen species (ROS, up to 0.1 mM). The authors designed a system of ROS-sensitive microbeads with alginate/polyvinyl alcohol (PVA)-based 1,4-phenyleneboronic acid (Pa) embedded in a gelatin-based gel (Figure 4A). The results showed an encapsulation efficiency above 95%, nontoxicity, and the ROS-triggered release of highly active *Lactobacillus plantarum* in intestinal inflammatory sites (Figure 4B). This effect occurred due to reversible reactions formed between PVA (1,2 diol free groups) and Pa (compounds derived from boronic acid and boronate esters) which, in turn, when they come into contact with hydrogen peroxide, are capable of breaking down and effectively releasing the active compound (Figure 4C) [130]. Similarly, polydopamine and catalase, considered powerful antioxidants against free radicals, were used to fabricate ROS scavenging microparticles via coprecipitation and self-polymerization. Dextran sulfate was further applied to modify the microparticles’ surface considering its targeting ability toward activated macrophages. Results showed the preparation of 1.1 µm microparticles with excellent ROS scavenging ability in cell experiments and a 40% reduction in disease activity index scores in animal experiments (Figure 4D) [131].

In fact, multicomponent microencapsulation could generate better results, as reported by Gąsiorowska et al. The authors showed a clinical study using sodium butyrate microparticles, probiotics, and short-chain fructooligosaccharides in patients with Irritable Bowel Syndrome (IBS), a functional gastrointestinal disorder that can cause symptoms similar to IBD but without damaging the gastrointestinal tract [132]. Probiotics used were two Lactobacillus strains (*L. rhamnosus* and *L. acidophilus*) and three *Bifidobacterium* strains (*B. longum*, *B. bifidum*, and *B. lactis*), and results showed that this combination could be beneficial for an alternative treatment of IBD, as it restores the disturbed functionality of the intestinal microbiota [115].

Microparticles in general present a high rate of protection of bioactive compounds; however, a small premature release of the drug was observed in the stomach, compromising the release in the colon for local treatment. Therefore, a complementary coating alternative was applied to overcome this challenge and optimize the system’s effectiveness [133]. Enteric coatings are widely used in the pharmaceutical industry to protect drugs from being broken down by stomach acid before they can be absorbed in the small intestine. Commonly used for active agents that are sensitive to the acidic environment of the stomach, such as probiotics and some antibiotics, enteric coatings can help improve their effectiveness and reduce the risk of side effects. Among the main polymers explored for such applications are cellulose acetate phthalate, polyvinyl acetate phthalate, and methacrylic acid copolymers [134]. Cellulose derivatives are widely used for this purpose, since their structure allows one to obtain rigid, plasticized, and pH-responsible structures [135,136].

Several polymers have been approved by the FDA for this technological process, but some are still under investigation. Park et al. (2016) coated microparticles containing the probiotics *Lactobacillus acidophilus* and *Bifidobacteria animalis* ssp. *Lactis* with hydroxypropyl methylcellulose acetate succinate (HPMCAS) using a dry powder coating technique. The preparation process did not affect the growth of the probiotics, which was improved in acidic conditions compared with encapsulated and uncoated bacteria. As a result, coated probiotics were able to colonize the small and large intestinal membrane, providing pharmacological advantages in relation to the free probiotics [137].

The probiotic *Lactobacillus acidophilus* and prebiotic Reishi medicinal mushroom (*Ganoderma lingzhi*) extract were co-encapsulated in calcium-alginate microparticles aiming to prolong their stability in gastric conditions. Microparticles were prepared with an ionotropic gelation technique using calcium lactate as a crosslinker agent. As a strategy to increase the stability of encapsulated agents, microparticles were double-coated with the same polyelectrolyte layer and a second crosslinking solution (6% calcium lactate + 0.4% chitosan). Single-layered and double-layered microparticles were 1.5 mm and 2.3 mm in diameter and showed 83% and 74% of encapsulation efficiency, respectively. The number of viable bacteria was higher for double-layer coating microparticles in relation to single-layer-ones, demonstrating the influence of coatings on reducing the release rate of the active molecules in harsh conditions of the upper GIT [138].

Another designed microsystem was using double-layer gelatin-mucin-alginate and chitosan-mucin-alginate to protect *Lactobacillus plantarum* B2, responsible for riboflavin production. The authors showed that the inclusion of mucin in the formulation made it possible to maintain the structural characteristics of the microcapsules even after lyophilization, in addition to increasing the rate of adherence in the intestinal tract and stimulating greater production of vitamin B2 [139]. Curiously, the application of mucin in other biomaterials has been shown to generate a positive immunological response for the patient since they would have been related to the expression of cytokines [140]. Likewise, Eudragit^®^ S100 (anionic polymer) is used as an enteric coating which, in the majority of its formulations, allows for the blocking of the release of biomolecules in acidic pH; as such, after the microencapsulation of *L. acidophilus* and the coating (chitosan nanoparticles and Eu S100) were incorporated into Iranian Doogh drinks, they showed that the storage time would be up to 42 days without changes in their morphology, preserving their organoleptic properties and their bacterial viability [141].

The structural modification of polymers can improve their biophysical properties of both release and adherence to the intestinal mucosa since there are some reports indicating the use of thiol groups as potential mucoadhesive agents [142,143]. Zhang et al. developed a multinucleated bilayer microsystem of thiolated carboxymethyl cellulose, allowing for greater adherence between probiotics (*Bifidobacterium adolescence* FS2-3 and *Bacillus subtilis* SN15-2) and the mucosa; significant repairs were observed in the microbiota and the site of inflammation induced by *Escherichia coli* O157:H7 [144]. On the other hand, some lipid supplements can be added to these microsystems to improve their versatility. A double emulsion of *L. rhamnosus* and krill oil based on protein showed a high content of monounsaturated and polyunsaturated fatty acids capable of avoiding the appearance of neurodegenerative diseases, obesity, and cancer, which lead to a multi-objective application of prevention [145]. Various emulsions were generated during recent years where probiotics were incorporated, but, curiously, the use of Okara oil allowed for the modulation and regulation of the size of the microcapsule structure since the size would be related to the concentration of the oily phase [146]. However, it is necessary to be careful at the time of formulation as there may be an induced stress which would allow some bacteria, such as *Lactobacillus plantarum*, when combined with Omega 3, to lose the anticancer activity that each component possessed individually [147].

A recent study indicated that the incorporation of *Akkermansia muciniphila* would be beneficial to prevent some diseases, including IBD, since a study in mice indicated that the counts of these bacteria in high-fat diets were significantly lower than those in healthy mice. Thus, Marcial-Coba et al., reported a study where *A. muciniphila* and *Lactobacillus* cases were included in xanthan/gellan gum microcapsules and incorporated into dark chocolate; they achieved gastric protection, a long storage period, high bacterial viability, and no significant difference in its organoleptic properties [148]. *Akkermansia* spp. is a bacterial family that is difficult to isolate and manipulate, and there is little information about its role in the microbiota as well as its role in the appearance of gastrointestinal diseases reported to date.

### 5.2. Nanoparticle-Based Drug Delivery Systems

With the advent of nanotechnology, it was possible to explore new delivery platforms that were able to solve problems related to the low stability and bioavailability of probiotics given orally. The encapsulation of probiotics in nanosystems can protect them against degradation, improving their stability during product processing and digestion, which improves their bioavailability in the large intestine by delivering a higher fraction of viable probiotics to the colon. In this session, the biological properties of nanosystems containing probiotics will be discussed as a therapeutic alternative for the management of IBD [149,150].

To ensure the therapeutic efficacy of probiotics, they must remain viable and intact within their formulation and resist various pH fluctuations during transit through different regions of the GIT, including the oral cavity, stomach, and intestines. Additionally, these probiotics must be able to withstand the effects of digestive enzymes, such as amylase, pepsin, and trypsin. This is crucial, as oral administration requires that the probiotics survive the harsh environment of the stomach and be released in the colon. Polymeric nanoparticles have proven to be particularly useful in the transport and delivery of probiotics as they can be designed from natural or synthetic polymers having some of the peculiarities of the colon as a trigger mechanism for release. Among them, the characteristics most explored to initiate the release are the distinct pH range (such as sodium-alginate- and Eudragit^®^-based systems), the specific enzymatic biodegradability (resistant starch-based systems), as well as the longer transit time (ethyl cellulose coating). The latter can be exploited by systems based on polymers that have poor aqueous solubility and that require longer contact time with gastrointestinal fluids for release [151,152].

*Bacillus amyloliquefaciens* loaded nanoparticles (BANPs) were designed to achieve a long-lasting effect in the prevention and treatment of IBD. The chitosan nanoparticles were more stable in simulated intestinal and gastric juice than free bacillus. After colitis induction, rats treated with BANPs had a higher survival rate and greater recovery of liver and kidney function. Fecal Lcn-2 levels were significantly decreased in rats treated with BANPs compared with the colitic group. The BANP group also showed reconstructed colon epithelium in some areas with reduced inflammatory cell infiltration and detachment of the epithelial lining. The study suggests that orally administered BANPs have potential therapeutic effects on colitis [153].

In a similar study, *B. amyloliquefacies* was associated with *Lactobacillus acidophilus* and *Bifidobacterium bifidum* in chitosan nanoparticles. Similar to the previous study, the nanoparticle system improved the stability of probiotics in the GIT and reduced the severity of clinical signs in colitic rats, including weight loss, diarrhea, and rectal bleeding. Additionally, the nanoparticle system reduced inflammatory markers and increased the expression of anti-inflammatory cytokines in colon tissues. The study suggests that the nanoparticle system has potential as a treatment for colitis [154].

Ebrahimnejad and coworkers (2017) developed chitosan-based nanoparticles for the encapsulation of *L. acidophilus* (1643 PTCC) as a probiotic agent. The polymeric nanoparticles were obtained using chitosan and tripolyphosphate anions via an ionic gelation technique and presented an average size of 100 to 250 nm with a distinct spherical regular shape, as confirmed with scanning electron microscopy. The optimal bacterial loading was achieved using the lowest chitosan concentration (0.05 mg/mL). The authors assessed the survival of *L. acidophilus* in vitro using simulated gastric fluid (pH 2), and results showed a better protective effect in encapsulated bacteria compared with its free form, corroborating the results described in the literature, where probiotic encapsulation seems to increase bacterial viability after exposure to the acidic conditions of the stomach. In intestinal conditions, the number of free cells remarkably decreased after 120 min, while encapsulated bacteria did not. The results suggest the encapsulation of probiotics using chitosan can improve the stability of probiotics in different pH conditions, improving their delivery to the colon [155].

Biodegradable chitosan-coated PLGA nanoparticles (NPs) for the delivery of probiotic *Lactobacillus plantarum* extract (LPE) to treat colitis were developed by Saadatzadeh et al. The NPs show a particle size of 226.3 nm, positive surface zeta potential, and biphasic controlled drug release. Treatment with LPE and NPs reduced colonic damage, decreased levels of inflammatory cytokines TNF-α and IL-1β, and reduced myeloperoxidase and lipid peroxidation activity in colitic rats. The study concluded that the LPE-loaded NPs can be a promising approach for treating colitis [156]. These studies suggest that chitosan-based nanoparticle systems have potential as a treatment for colitis by improving the stability and delivery of probiotics.

Sahu and coauthors explored agricultural waste as an opportunity to design eggshell nanoparticle-based (ESNP) gel for probiotic delivery. ESNPs were prepared using sterile air-dried eggshells that were processed in a ball mill. The ESNPs presented a size <300 nm. The gel was prepared using a suspension of *L. plantarum* (MTCC2621) and the ESNPs. After incorporation, the gel presented 80% of the initial *L. plantarum* content. This concentration decreased by 40% in the gel after 60 days of storage, while 100% of cells were lost after 60 days in the conventional formulation, suggesting a protective effect of ESNP gel. The stability in simulated gastric fluid was assessed using simulated gastric fluid prepared with pepsin and other salts at pH 2. The gel did not present a reduction in viability even after 120 min, while for free cells a reduction of 50% was observed. These results reinforce the ability of ESNP-based gels to improve probiotic delivery by shielding the bacteria from the acidic pH of the gastric fluid and their potential use in the management of intestinal diseases [157].

Self-assembling hyaluronic acid (HA)-based nanoparticles capable of scavenging reactive oxygen species (ROS) conjugated with probiotic *Escherichia coli* Nissle 1917 (EcN) within a nanogel layer containing hinokitiol (HPN), a natural anti-inflammatory compound for targeted delivery to inflamed colon tissues, were developed by Liu et al. The nanoparticles effectively protect the EcN against environmental assaults, such as simulated gastric fluid and bile salts, and exhibited enhanced survival rates compared with uncoated EcN. The conjugation of HPN onto the surface of EcN has the potential to exhibit synergy for enhanced therapeutic efficacy in IBDs. The treatment was tested on mice with induced colitis and was found to significantly reduce weight loss, increase colon length, and minimize colon damage compared with other treatments. The therapy also modulated the gut microbiota by increasing the relative abundance of beneficial bacteria and decreasing virulent bacteria [158].

Regarding inorganic nanoparticles, their use for food and medical applications is limited due to issues related to their low solubility and intrinsic toxicity resulting from the use of heavy metals in their fabrication. Furthermore, when compared to other nano-fabrication techniques, inorganic nanoparticles do not provide significant advantages in protecting probiotics from degradation.

Interestingly, Wei and colleagues explored one possible use of mesoporous silica nanoparticles (MSNPs) to create a multi-purpose nanoprobiotic. The authors encapsulated *B. infantis* (ATCC 15697) with MSNPs, which were further coated with bacterially derived quantum dots (QDs) produced either by *L. acidophilus* or *E. coli*. This multi-purpose nanoprobiotic (MPNPs) was successfully fabricated, as techniques such as infra-red spectroscopy and scanning electron microscopy confirmed the MSNP coating in *B. infantis* and the further attachment of bacterially derived QDs on its surface. In vitro exposure of MPNPs to simulated gastric fluid and antibiotics demonstrated better protection compared with uncoated *B. infantis*, suggesting that an MSNP coating could be useful to protect probiotics from gastric fluid. The MPNPs also showed increased mucoadhesion on an intestinal epithelial layer colonized by *E. coli* when compared with uncoated *B. infantis*, suggesting a possible therapeutic effect against Enterobacteriaceae colonization. Additionally, the QD-coated nanoprobiotics showed an increased killing effect on *E. coli* due to the increased generations of reactive oxygen species. These techniques together could lead to an enhancement of the properties of *B. infantis*, representing a new generation of engineered nanoprobiotics [159].

Probiotic-inspired nanomedicines were developed by coating MSNP with an *Escherichia coli* Nissle 1917-derived membrane. The researchers found that the nanoparticles were effective in treating IBDs by regulating the redox balance and immune responses in a mouse model of colitis. Compared with other nanoformulations and conventional therapeutics, the nanoparticles demonstrated the greatest therapeutic effects, including reduced weight loss, a lower disease activity index (DAI), increased colon length, and decreased colonic histological damage scores. The treatment with the nanoparticles also improved the integrity of the colonic mucosal barrier, as indicated by increased levels of tight junction markers. It is suggested that the adhesion of the membrane shell to the mucus layer and the accumulation of the nanoparticles in inflammatory tissue contributed to its therapeutic efficacy. The NP was found to decrease myeloperoxidase (MPO) activity and pro-inflammatory cytokine secretion while increasing the secretion of the anti-inflammatory factor interleukin-10 (IL-10) in the colon tissue of acute colitis mice [160].

In recent decades, research related to the use of lipid nanoparticles for the delivery of probiotics has gained prominence with the advent of nanocoating techniques that have improved the biopharmaceutical properties of colloidal lipid carriers such as liposomes. Polysaccharides and protein-coated liposomes are capable of protecting probiotics against degradation, significantly increasing the number of viable probiotics after exposure to simulated gastric juice and simulated intestinal juice, reinforcing their potential application in the delivery of probiotics to the intestine [152,161,162].

The use of nanotechnology has provided a promising strategy for the delivery of probiotics in the treatment of IBD. The encapsulation of probiotics in nanosystems protects them from degradation, improves their stability and bioavailability, and enhances their therapeutic efficacy. Polymeric nanoparticles, particularly those designed to withstand the harsh conditions of the GIT, have shown potential in the targeted delivery of probiotics to the colon in addition to providing bioadhesive properties for the formulations. Several studies have demonstrated the benefits of probiotic-loaded nanoparticles in reducing the severity of clinical signs of IBD and restoring the epithelial lining in the colon [163]. These findings indicate that nanotechnology-based approaches have the potential to improve the effectiveness of probiotics in the treatment of intestinal inflammatory diseases.

## 6. Conclusions

This review underscores the immense promise of probiotics delivered through micro- and nanoparticles as a novel strategy for managing IBD. While substantial progress has been made in understanding the potential benefits and mechanisms of probiotic-loaded particles, numerous avenues for future research and exploration remain. To pave the way for more effective IBD treatments, it is imperative that further investigation and research are undertaken.

Future research in this field should focus on optimizing the formulation of probiotic-loaded particles. This includes developing more advanced and targeted delivery systems, enhancing the stability of probiotics during transit through the GIT, and exploring innovative encapsulation methods to ensure prolonged probiotic viability. Moreover, personalized medicine holds significant potential. Investigating the tailoring of probiotic therapies to individual patient profiles based on their unique gut microbiota composition can provide more personalized and effective treatment regimens. Longitudinal studies and clinical trials are required to assess the sustained safety, efficacy, and potential long-term side effects of probiotic-loaded particles for IBD. Additionally, a deeper understanding of the immunomodulatory mechanisms and the interactions between probiotics and the host’s immune system in the context of IBD is essential for refining treatment strategies. This would involve exploring the specific strains of probiotics and their dosages that yield the most favorable outcomes for various IBD subtypes.

The potential future research directions in the field of IBD treatment with probiotics in micro- and nanoparticles are vast. These investigations will not only enhance our comprehension of the therapeutic potential of probiotics but also offer innovative, patient-specific solutions for managing this challenging and prevalent disease. It is with these prospects in mind that we look forward to the continued evolution of IBD therapeutics and the improved quality of life for patients affected by this condition.

## Figures and Tables

**Figure 1 pharmaceutics-15-02600-f001:**
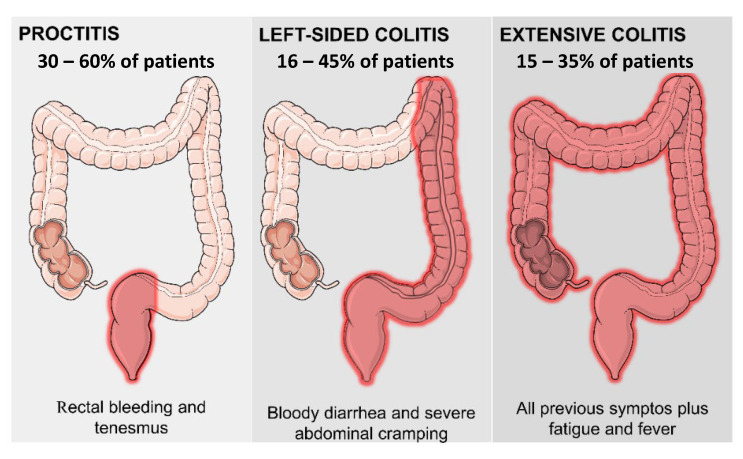
Ulcerative colitis classification with incidence and main symptoms.

**Figure 2 pharmaceutics-15-02600-f002:**
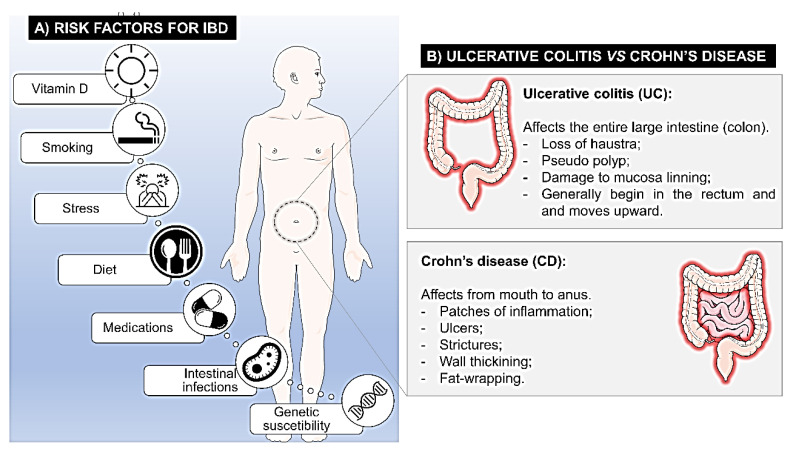
(**A**) Factors of risk for IBD and (**B**) main differences between ulcerative colitis and Crohn’s disease.

**Figure 3 pharmaceutics-15-02600-f003:**
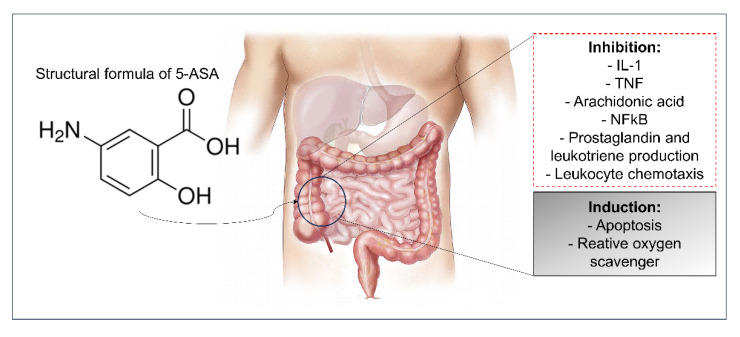
Structural formula and mechanism of action of 5-ASA.

**Figure 4 pharmaceutics-15-02600-f004:**
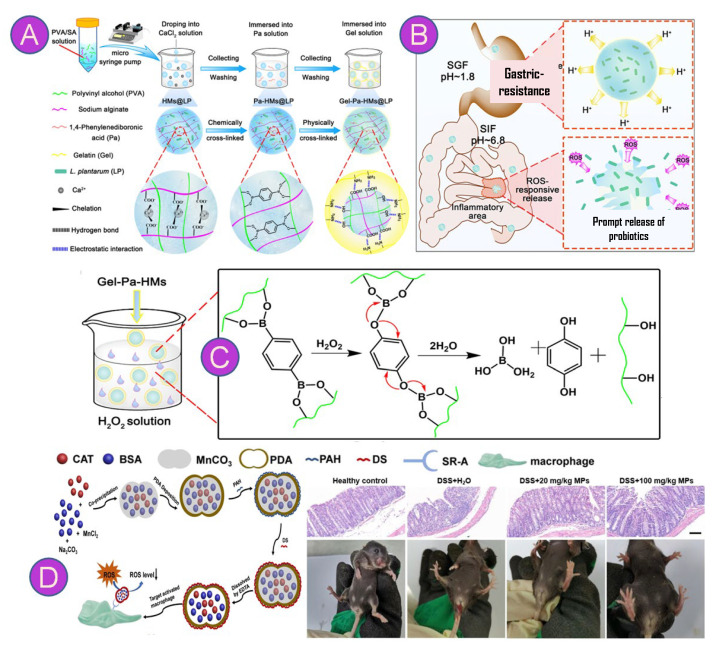
(**A**) Schematic illustration of the fabrication and mechanism of the Gel-Pa-HMs@LP. (**B**) Schematic illustration of the encapsulated *Lactobacillus plantarum* precisely delivered to the inflammatory area and released in response to ROS. (**C**). Schematic illustration of the ROS scavenging mechanism of the Gel-Pa-HMs. Reprinted (adapted) with permission from Zhu et al. [129]. Copyright 2023 American Chemical Society. (**D**) H&E-stained pathological sections of colons of mice with different treatments on day 6, scale bar = 100 μm, and representative photographs of rectal areas of a healthy mouse and IBD mice after treatment with H_2_O and 20 and 100 mg/kg microparticles. Reprinted (adapted) with permission from Li et al. [131]. Copyright 2021 American Chemical Society.

**Table 1 pharmaceutics-15-02600-t001:** Benefits of the use of probiotics, alone or combined with 5-ASA, in the treatment of IBD.

Microorganisms	Benefit	Reference
*Lactobacillus* spp. and *Bifidobacterium* spp.	Inflammation reduction	[14]
*Lactobacillus* and *Bifidobacterium* plus *Streptococcus* or *E. coli* Nissle 1917	Symptom remission	[15]
*Lactobacillus* spp., *Bifidobacterium* spp., and *Streptococcus thermophilus*	Symptom attenuation	[16]
*Saccharomyces boulardii*	Angiogenesis blockage	[13]
*Bacillus*	Survive in acid conditions	[17]
*Saccharomyces boulardii, Akkermansia muciniphila*, and *Faecalibacterium prausnitzi*	Intestinal barrier reinforcement; colonic inflammation reduction	[19]
*Enterococcus faecium* CRL 183 and *Bifidobacterium longum* ATCC 15707	Inflammation and ulceration reduction; *Lactobacillus* spp. and *Bifidobacterium* spp. population increase in rat colon	[12]
5-ASA plus L. *salivarius*, L. *acidophilus*, and *B. bifidum* BGN4	Recovery time reduction; symptom attenuation	[18]

## Data Availability

The data presented in this study are available in this article.

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
