# Peer review of "Delivery Strategies of Probiotics from Nano- and Microparticles: Trends in the Treatment of Inflammatory Bowel Disease—An Overview"

_pharmaceutics, 2023, doi:10.3390/pharmaceutics15112600_

Round 1

Reviewer 1 Report

Comments and Suggestions for Authors

The present manuscript is an overview of the immunological implications of inflammatory bowel disease, current therapeutic options, effect of probiotics and micro-/nanocarriers for delivery of the latter to the colon.

Although the topics are of high current interest, in my opinion the manuscript presents the shortcomings listed below.

-The title of the manuscript does not fully reflect its contents. It is not clearly indicate that only micro-/nanocarrier formulations are reviewed. As such, it may be misleading, probiotic delivery systems other than micro-/nanocarriers being described in the literature.

-The structure of the article is questionable. The Introduction is mostly a preview of the subsequent sections. The anatomy section is mainly common knowledge and would not be necessary, whereas the immunology one is quite in depth and very poorly related to the drug delivery part. Thus, the overall text appears somewhat disjointed.

-The drug delivery section lacks clarity, due to poorly fluent wording and rough description of many delivery systems reviewed. This section should be written more clearly and broadened to be the focus of the article.

-Specific comments:

The abstract would not comply with the Author guidelines, as the text is divided into subheadings.

The Authors should use “Inflammatory Bowel Disease” rather than “Inflammatory Bowel Diseases”. Indeed, according to Centers for Disease Control and Prevention (CDC) and most other authoritative sources, Inflammatory Bowel Disease is a term for two conditions (Crohn’s disease and ulcerative colitis) that are characterized by chronic inflammation of the gastrointestinal (GI) tract.

Line 22. With prebiotics, it would more be a matter of damage and loss of viability than of degradation.

Line 46. Please check “psoriasis”.

Lines 72-74. This passage is quite confusing: modified-release and prodrug approaches are mixed up, and different modified-release products are mentioned altogether. Also, no time-dependent formulations are available on the market.

Line 137. Please check “zonules occluding”.

Line 138. What do the Authors mean by “the distal portion of the proximal intestine (beginning of the small intestine)”?

Line 165. Any connections between intestinal microbiota and brain functions (gut-brain axis), if mentioned, should be better explained.

Line 193 (and throughout the manuscript). Although some acronyms may be known, the full expression they refer to should be indicated on their first mention.

Lines 195-196. This is well known, no consequence of concepts reported in the previous passages.

Line 110. Please check “phagocytosed”. Should it read phagocytize?

Line 317. Please check the hierarchy of this heading (§5).

Figure 2. Is this image original or is it drawn from previous reports? In the latter case, please indicate source. In addition, it is suggested the different types of ulcerative colitis be named as in the text.

Lines 552-555. The range of pH values encountered throughout the gastrointestinal tract would not exactly be as depicted. Moreover, it would not represent the main challenge to oral colonic release. The actual challenge is to shield the drug in the stomach and small intestine then making it selectively available in the colon.

Line 599. Please check “anti-inflammatory responses”.

Line 668. “reduces” should be changed to “reduce”.

Line 685. “evaluated” should be changed to “evaluate”.

Lines 745-747. The concepts of survival and viability would not apply to drug molecules.

Figure 4. Sketch in panel B is unclear and should be redrawn or deleted.

Line 791. Irritable bowel syndrome (IBS) is other than IBD.

Lines 889-890. To ensure protection throughout the upper gastrointestinal tract, not only should a material be resistant to enzymatic breakdown but also poorly water soluble and permeable. This would not be the case with some of the polysaccharides that are here mentioned.

Lines 898-899. What do the Authors mean by “recovered liver and kidney function tests after colitis induction”?

Line 927. “shows” should be changed to “show”.

Line 980-981. What do the Authors mean by “MSN coated with the membrane of a probiotic bacterial strain called Escherichia coli Nissle 1917 (EcN) was developed by Zu et al”?

Author Response

Reviewer 1

Inflammatory bowel disease, current therapeutic options, effect of probiotics and micro-/nanocarriers for delivery of the latter to the colon.

Although the topics are of high current interest, in my opinion the manuscript presents the shortcomings listed below.

-The title of the manuscript does not fully reflect its contents. It is not clearly indicate that only micro-/nanocarrier formulations are reviewed. As such, it may be misleading, probiotic delivery systems other than micro-/nanocarriers are being described in the literature.

We appreciate the reviewer suggestion. Thus, the title was changed to “Delivery Strategies of Probiotics-loaded Nano and Microparticles: Trends for the Treatment of Inflammatory Bowel Disease – An Overview”

-The structure of the article is questionable. The Introduction is mostly a preview of the subsequent sections. The anatomy section is mainly common knowledge and would not be necessary, whereas the immunology one is quite in depth and very poorly related to the drug delivery part. Thus, the overall text appears somewhat disjointed.

The Introduction section was restructured, and the anatomy section removed, as indicated. The socio-economic consequences information also was removed, as suggested by the second reviewer. 

Reviewing the immunology section, we came to the same conclusion that in fact it is in depth and poorly related to the drug delivery. Since the Immunopathogenesis section already talks on immunology, an important aspect of IBD, we agree with the reviewer and removed the immunology section.

-The drug delivery section lacks clarity, due to poorly fluent wording and rough description of many delivery systems reviewed. This section should be written more clearly and broadened to be the focus of the article.

The text was corrected.

-Specific comments:

The abstract would not comply with the Author guidelines, as the text is divided into subheadings.

Correction made.

The Authors should use “Inflammatory Bowel Disease” rather than “Inflammatory Bowel Diseases”. Indeed, according to Centers for Disease Control and Prevention (CDC) and most other authoritative sources, Inflammatory Bowel Disease is a term for two conditions (Crohn’s disease and ulcerative colitis) that are characterized by chronic inflammation of the gastrointestinal (GI) tract.

Correction made.

Line 22. With prebiotics, it would more be a matter of damage and loss of viability than of degradation.

Correction made.

Line 46. Please check “psoriasis”.

It was removed.

Lines 72-74. This passage is quite confusing: modified-release and prodrug approaches are mixed up, and different modified-release products are mentioned altogether. Also, no time-dependent formulations are available on the market.

The paragraph was rewritten. As indicated, various dosage forms with distinct system of delivery are available in the market, including pH-dependent, time-dependent and those activated by bacteria digestion. Pentasa is a classical example of time-dependent release characteristics (Zhang M, Merlin D. Nanoparticle-Based Oral Drug Delivery Systems Targeting the Colon for Treatment of Ulcerative Colitis. Inflamm Bowel Dis [Internet]. 2018 Jun 8;24(7):1401–15. Available from: https://academic.oup.com/ibdjournal/article/24/7/1401/4999330)

Line 137. Please check “zonules occluding”.

This section (Anatomy of GIT) was removed as suggested.

Line 138. What do the Authors mean by “the distal portion of the proximal intestine (beginning of the small intestine)”?

The section containing this paragraph has been removed.

Line 165. Any connections between intestinal microbiota and brain functions (gut-brain axis), if mentioned, should be better explained.

Correction made in the paragraph (line 375).

Line 193 (and throughout the manuscript). Although some acronyms may be known, the full expression they refer to should be indicated on their first mention.

Corrections made.

Lines 195-196. This is well known, no consequence of concepts reported in the previous passages.

Correction made.

Line 210. Please check “phagocytosed”. Should it read phagocytize?

In fact, should be “phagocytized”. Correction made.

Line 317. Please check the hierarchy of this heading (§5).

Checked. No changes needed.

Figure 2. Is this image original or is it drawn from previous reports? In the latter case, please indicate source. In addition, it is suggested the different types of ulcerative colitis be named as in the text.

The image is original. Correction made.

Lines 552-555. The range of pH values encountered throughout the gastrointestinal tract would not exactly be as depicted. Moreover, it would not represent the main challenge to oral colonic release. The actual challenge is to shield the drug in the stomach and small intestine then making it selectively available in the colon.

This paragraph was removed

Line 599. Please check “anti-inflammatory responses”.

Checked. The information is corrected, as can be seen in “Khan I, Ullah N, Zha L, Bai Y, Khan A, Zhao T, Che T, Zhang C. Alteration of Gut Microbiota in Inflammatory Bowel Disease (IBD): Cause or Consequence? IBD Treatment Targeting the Gut Microbiome. Pathogens. 2019 Aug 13;8(3):126.

Line 668. “reduces” should be changed to “reduce”.

Correction made.

Line 685. “evaluated” should be changed to “evaluate”.

Correction made (line 504)

Lines 745-747. The concepts of survival and viability would not apply to drug molecules.

Corrected in terms of drug stability and probiotics viability. Line 751

Figure 4. Sketch in panel B is unclear and should be redrawn or deleted.

The Figure was modified.

Line 791. Irritable bowel syndrome (IBS) is other than IBD.

The text was complemented by bringing the difference between IBD and IBS.

Lines 889-890. To ensure protection throughout the upper gastrointestinal tract, not only should a material be resistant to enzymatic breakdown but also poorly water soluble and permeable. This would not be the case with some of the polysaccharides that are here mentioned.

The text has been corrected, highlighting the main characteristics of the colon that can be exploited as trigger for colon delivery (from 890)

Lines 898-899. What do the Authors mean by “recovered liver and kidney function tests after colitis induction”?

Correction made (line 901)

Line 927. “shows” should be changed to “show”.

Correction made. All verbs in the text were checked. 

Line 980-981. What do the Authors mean by “MSN coated with the membrane of a probiotic bacterial strain called Escherichia coli Nissle 1917 (EcN) was developed by Zu et al?

The text has been corrected (line 984). This is a nanomedicine inspired by probiotics, and not the use of a probiotic itself.

Reviewer 2 Report

Comments and Suggestions for Authors

The title of manuscript “Trends for the treatment of Inflammatory Bowel Diseases: Immunological Implications and Delivery Strategies of Probiotics - An Overview” submitted by Sílvio André Lopes and colleagues, suggests a critical analysis of the issue. However, the "Introduction", instead of posing problems related to the topic of the article, sets out the characteristics of the disease and its socio-economic consequences, using not always recent publications (for example, lines 67-70; 79-83; 92-94, etc.).  Questions arose while reading subsequent sections.

Section 2 “Anatomy of the GIT” is puzzling: why present generally known information using publications before 2021? I suspect that potential readers of this review know how the intestines are structured and work...

Section 3 “The gut microbiota” very briefly presents general information about the set of microorganisms inhabiting the gastrointestinal tract, data from publications 2016-2021. Based on the goals of the Review, this section should undoubtedly be more detailed and provide current information about the microbiota in IBD. According to PubMed data, 984 articles on the study of the microbiome in IBD were published in 2022, and more than 700 in 2023.

Section 4. “The intestinal immune system” is written based on publications from 2013-2015 and contains only one reference to work published in 2021. Considering the rapid development of immunological research methods, the information provided is obviously outdated.

Section 5 “Inflammatory Bowel Disease (IBD)” is disappointing because it is also written on the basis of outdated data, especially the subsections 5.2.1; 5.2.2 and 5.2.3. Subsection “5.3. Immunopathogenesis of IBD,” which is the “main” section for this review, also does not contain new publications.

Section 6. “Treatment of IBD” is disappointing: nothing new has appeared in the treatment of IBD over the past 5 years... Section 7. “Dysbiosis and the use of probiotics in the treatment of IBD” also contains no new data.

Finally, Section 8 “Drug and Probiotic Delivery Systems.” The part describing the use of microcapsules presents the material clearly, mainly using publications from 2021-2023. The section describes a number of types of microparticles and their uses for drug and probiotic delivery.

Regarding the delivery of probiotics using nanoparticles, in my opinion, the formulation of the question does not correspond to the definition of nanoparticles. Nanoparticles, as accepted by the international community, cannot be larger than 100 nm in at least one dimension. Accordingly, nanoparticles cannot contain bacteria whose size exceeds 0.5 microns. How can nanoparticles encapsulate bacteria?

Nevertheless, the information provided by the authors on the use of nanosystems for the delivery of probiotics is interesting and this direction is, without a doubt, is developing. It is known that chitosan and other polymers used in nanotechnology can also form microparticles/microcapsules. I advise authors to use adequate terms, it will show that authors understand the sense of the issue. In any case, I believe that accuracy of definitions is necessary and useful.

The content of Section 8 shows the state of probiotic delivery research in general, but may be supplemented by the results of recent research.

The “Review” genre involves not just a brief retelling of articles on any topic, but a critical analysis of their results, generalization of an array of data and identification of promising directions for the development of the area being analyzed. Undoubtedly, the “Review” should cover research of recent years, if it is not of a historical nature.

The work “Trends for the treatment of Inflammatory Bowel Diseases: Immunological Implications and Delivery Strategies of Probiotics - An Overview”, unfortunately, does not have the characteristics of a “Review”.

If we talk about reworking the review, then the first 6 sections should be reduced; think through sections 7-8 taking into account publications 2021-23 and formulate the “trends” stated in the title. The current version is not acceptable for publication.

Comments on the Quality of English Language

Overall the text is written in clear language, however editing by a native English speaker would be helpful

Author Response

Reviewer 2

The title of manuscript “Trends for the treatment of Inflammatory Bowel Diseases: Immunological Implications and Delivery Strategies of Probiotics - An Overview” submitted by Sílvio André Lopes and colleagues, suggests a critical analysis of the issue.

However, the "Introduction", instead of posing problems related to the topic of the article, sets out the characteristics of the disease and its socio-economic consequences, using not always recent publications (for example, lines 67-70; 79-83; 92-94, etc.).  Questions arose while reading subsequent sections.

Section 2 “Anatomy of the GIT” is puzzling: why present generally known information using publications before 2021? I suspect that potential readers of this review know how the intestines are structured and work...

We accepted Reviewer suggestion and Section 2 was removed.

Section 3 “The gut microbiota” very briefly presents general information about the set of microorganisms inhabiting the gastrointestinal tract, data from publications 2016-2021. Based on the goals of the Review, this section should undoubtedly be more detailed and provide current information about the microbiota in IBD. According to PubMed data, 984 articles on the study of the microbiome in IBD were published in 2022, and more than 700 in 2023.

New and more recent data were added.

Section 4. “The intestinal immune system” is written based on publications from 2013-2015 and contains only one reference to work published in 2021. Considering the rapid development of immunological research methods, the information provided is obviously outdated.

Following considerations posed by Reviewer 1, the intestinal immune system section was removed.

Section 5 “Inflammatory Bowel Disease (IBD)” is disappointing because it is also written on the basis of outdated data, especially the subsections 5.2.1; 5.2.2 and 5.2.3. Subsection “5.3. Immunopathogenesis of IBD,” which is the “main” section for this review, also does not contain new publications.

Correction made.

Section 6. “Treatment of IBD” is disappointing: nothing new has appeared in the treatment of IBD over the past 5 years...

Correction made.

Section 7. “Dysbiosis and the use of probiotics in the treatment of IBD” also contains no new data.

Correction made.

Finally, Section 8 “Drug and Probiotic Delivery Systems.” The part describing the use of microcapsules presents the material clearly, mainly using publications from 2021-2023. The section describes a number of types of microparticles and their uses for drug and probiotic delivery.

Regarding the delivery of probiotics using nanoparticles, in my opinion, the formulation of the question does not correspond to the definition of nanoparticles. Nanoparticles, as accepted by the international community, cannot be larger than 100 nm in at least one dimension. Accordingly, nanoparticles cannot contain bacteria whose size exceeds 0.5 microns. How can nanoparticles encapsulate bacteria?

In general, nanotechnology involves the manipulation of structures and materials at the nanoscale, which can vary in size from 1 to 100 nanometers. However, in pharmaceuticals sciences, nanotechnology typically deals with structures and materials in the range from 1 to 1000 nm (Vivek Verma, Kevin M. Ryan, Luis Padrela, Production and isolation of pharmaceutical drug nanoparticles,

International Journal of Pharmaceutics,

Volume 603, 2021,

120708,). These nanoscale materials are employed to optimize drug delivery systems, and address issues like improving drug solubility, targeting specific cells or tissues, and controlling release profiles.

Besides, there are many studies providing that probiotics nanoparticles approach is possible, as shown during the manuscript. Which is your opinion about it? Keep it ou skip it?

Nevertheless, the information provided by the authors on the use of nanosystems for the delivery of probiotics is interesting and this direction is, without a doubt, is developing. It is known that chitosan and other polymers used in nanotechnology can also form microparticles/microcapsules. I advise authors to use adequate terms, it will show that authors understand the sense of the issue. In any case, I believe that accuracy of definitions is necessary and useful.

We know about the differentiation of structures in matrix and reservoir-type systems, technically named as micro/nanospheres and micro/nanocapsules, respectively. Obtaining a specific type of particle depends not only on the polymer used, but also on the system preparation technique employed. Finally, we know that the classification of a micro/nanosphere particle or capsule is dependent on a series of characterization techniques. Therefore, the terms used to reference the systems were the same as those used by the authors, and in case of doubt, the general term micro/nanoparticle, which includes micro/nanospheres and micro/nanocapsule, was used.

The content of Section 8 shows the state of probiotic delivery research in general, but may be supplemented by the results of recent research.

Preciso de ajuda dos co-autores.

The “Review” genre involves not just a brief retelling of articles on any topic, but a critical analysis of their results, generalization of an array of data and identification of promising directions for the development of the area being analyzed. Undoubtedly, the “Review” should cover research of recent years, if it is not of a historical nature.

Correction made and about the remaining comments, the focus of this work is not to be so inflexible, putting the origin and the contextualizing about these points.

The work “Trends for the treatment of Inflammatory Bowel Diseases: Immunological Implications and Delivery Strategies of Probiotics - An Overview”, unfortunately, does not have the characteristics of a “Review”.

If we talk about reworking the review, then the first 6 sections should be reduced; think through sections 7-8 taking into account publications 2021-23 and formulate the “trends” stated in the title. The current version is not acceptable for publication.

Changes were made and sections 1 to 6 shortened, focusing on aspects of probiotic release.

Reviewer 3 Report

Comments and Suggestions for Authors

According to the Author's review article, IBD has caused immense suffering and economic costs to many people worldwide for over a century, and its exact cause remains unknown. Treatment involves constant oral administration of anti-inflammatories such as 5-ASA, which can have adverse side effects that may cause some patients to stop taking the medication. Furthermore, the drug's effectiveness is reduced due to degradation during its passage through the superior GIT. Several studies have shown that IBD is strongly associated with dysbiosis and that disease symptoms can be minimized through oral ingestion of probiotics. Combining 5-ASA and probiotics appears to be a new approach to treat IBD. Co-encapsulating the drug and probiotics in micro and nanoparticles can minimize degradation in the GIT, favoring delivery specifically into the colon. This alternative treatment represents a significant therapeutic advancement that can increase the anti-inflammatory action of 5-ASA while reducing adverse side effects and restoring the intestine microbiota, thus considerably improving the quality of life of affected patients.

Minors:

The review article is well structured, however it could be improved with the inclusion of new figures and tables.

The authors could provide the structural formula of 5-ASA.

The authors could schematically show the action of 5-ASA (similar to Figure 1), i.e., show the text from line 509 schematically.

The authors should tabulate the nanocarriers for easier viewing of the information, make a certain classification, and list the basic characteristics.

The authors could use a Figure to show the structures of applicable polymers.

Author Response

Reviewer 3

According to the Author's review article, IBD has caused immense suffering and economic costs to many people worldwide for over a century, and its exact cause remains unknown. Treatment involves constant oral administration of anti-inflammatories such as 5-ASA, which can have adverse side effects that may cause some patients to stop taking the medication. Furthermore, the drug's effectiveness is reduced due to degradation during its passage through the superior GIT. Several studies have shown that IBD is strongly associated with dysbiosis and that disease symptoms can be minimized through oral ingestion of probiotics. Combining 5-ASA and probiotics appears to be a new approach to treat IBD. Co-encapsulating the drug and probiotics in micro and nanoparticles can minimize degradation in the GIT, favoring delivery specifically into the colon. This alternative treatment represents a significant therapeutic advancement that can increase the anti-inflammatory action of 5-ASA while reducing adverse side effects and restoring the intestine microbiota, thus considerably improving the quality of life of affected patients.

Minors:

The review article is well structured, however it could be improved with the inclusion of new figures and tables.

The authors could provide the structural formula of 5-ASA.

Request was accepted.

The authors could schematically show the action of 5-ASA (similar to Figure 1), i.e., show the text from line 509 schematically.

The authors should tabulate the nanocarriers for easier viewing of the information, make a certain classification, and list the basic characteristics.

The authors could use a Figure to show the structures of applicable polymers.

A schematic illustration demonstrating the mechanism of action of 5-ASA along with its structural formula has been added.

We appreciate your suggestion to use a Figure to illustrate the structures of polymers for colon delivery of probiotics. However, after careful consideration, we believe that it may not be necessary for this particular research. Our study primarily focuses on the efficacy and bioavailability of probiotics in the colon, and we have chosen to emphasize this aspect rather than delving into the chemical structures of the polymers used, extensively explored in other review articles.

Reviewer 4 Report

Comments and Suggestions for Authors

This article reviews the immunological significance, delivery strategy and the latest research progress of probiotics targeting inflammatory bowel disease, to help readers better understand the therapeutic strategy of probiotics. This study is well organized and summarized. Therefore, I am in favor of acceptance after addressing the following questions.

My comments for the current manuscript are listed as follow:

1.     The review mentions several studies, but there are no specific citations or references to support the claims made. It's crucial to include proper citations to back up the information presented.

2.     Some sentences are incomplete or poorly structured. For example, "The literature on IBD, published mostly in the last decade, was reviewed." This sentence lacks clarity and detail.

3.     There are grammatical and punctuation issues in the text, such as inconsistent capitalization and sentence structure.

4.     The introduction should provide more context about the significance of IBD, its prevalence, and the current state of treatment options. It should also outline the goals of the review.

5.     The review mentions probiotics as a potential treatment for IBD but does not delve into specific probiotic strains or their mechanisms of action. More detailed information about the use of probiotics is necessary.

6.     The review does not conclude with a section on potential future research directions or areas where further investigation is needed.

7.     Some related work or reviews should be cited (e.g., Acta Biomaterialia, 2023, 160: 252-264; Materials & Design, 2022, 218: 110686; Advanced Functional Materials, 2023: 2300261; ACS Applied Materials & Interfaces, 2022, 14(45): 50677-50691).

8.     Some terms and concepts are mentioned without adequate explanation. For example, "micro and nanoencapsulation technologies" are introduced without providing a clear understanding of what they are and how they work.

Comments on the Quality of English Language

Good.

Author Response

Reviewer 4

  1. The review mentions several studies, but there are no specific citations or references to support the claims made. It's crucial to include proper citations to back up the information presented.

Proper citations were included as already suggested by other authors.

  1. Some sentences are incomplete or poorly structured. For example, "The literature on IBD, published mostly in the last decade, was reviewed." This sentence lacks clarity and detail.

We completely agree and have therefore restructured the summary to make it easier to understand.

  1. There are grammatical and punctuation issues in the text, such as inconsistent capitalization and sentence structure.

The entire text has been revised.

  1. The introduction should provide more context about the significance of IBD, its prevalence, and the current state of treatment options. It should also outline the goals of the review.

We agree with the comment and new information has been added to the introduction.

  1. The review mentions probiotics as a potential treatment for IBD but does not delve into specific probiotic strains or their mechanisms of action. More detailed information about the use of probiotics is necessary.

An entire section was devoted to the benefits and mechanisms of action of probiotics in the treatment of IBD.

  1. The review does not conclude with a section on potential future research directions or areas where further investigation is needed.

The conclusion of the article has been completely rewritten, including information on potential future research directions and areas where further investigation is needed.

  1. Some related work or reviews should be cited (e.g., Acta Biomaterialia, 2023, 160: 252-264; Materials & Design, 2022, 218: 110686; Advanced Functional Materials, 2023: 2300261; ACS Applied Materials & Interfaces, 2022, 14(45): 50677-50691).

References were included.

  1. Some terms and concepts are mentioned without adequate explanation. For example, "micro and nanoencapsulation technologies" are introduced without providing a clear understanding of what they are and how they work.

Writing about the methods of micro and nanoparticle preparation presents a unique set of challenges. These processes often involve intricate scientific techniques, precise equipment, and complex chemical or physical principles that can be challenging to convey in a clear manner in an article not intended to do so.

Round 2

Reviewer 1 Report

Comments and Suggestions for Authors

The Authors have made an effort to comply with the Reviewers’ suggestions.

Few minor comments are reported below.

- Line 52. I would use “release mechanisms” rather than “mechanisms of action” when referring to delivery systems.

- I do not agree about defining Pentasa and Mezavant as time-dependent and time-delayed, respectively. Pentasa is a prolonged-release dosage form having reduced rate of drug release. Mezavant is a pH-responsive dosage form, provided with an enteric coating and a prolonged-release drug-containing core. Conversely, “time-dependent” properly refers to delivery systems that release the drug only after a lag phase programmed to cover relatively consistent small intestinal transit time. The lag phase would start after gastric emptying thus allowing colon targeting based on the time-dependent approach. In this sense, Mezavant are quite different dosage forms that be defined as pH-responsive and prolonged-release, respectively.

Line 62. Please check “reach”

§ 5. Anti-inflammatory drug delivery does not seem to be covered in this section and is not even mentioned in the manuscript title. Thus, it could be left out from the heading and the introductory paragraph of the delivery section.

Line 631-32. Passage “Therefore, the use of the drug in its entirety is not allowed, which leads to opting for a complementary alternative, the coating process” is not fully clear and needs to be rephrased.

Line 651. “viability” would not apply to the prebiotic.

Author Response

REVIEWER 1

The Authors have made an effort to comply with the Reviewers’ suggestions.

Few minor comments are reported below.

- Line 52. I would use “release mechanisms” rather than “mechanisms of action” when referring to delivery systems.

The authors appreciate your comment. The word has been modified in the sentence.

- I do not agree about defining Pentasa and Mezavant as time-dependent and time-delayed, respectively. Pentasa is a prolonged-release dosage form having reduced rate of drug release. Mezavant is a pH-responsive dosage form, provided with an enteric coating and a prolonged-release drug-containing core. Conversely, “time-dependent” properly refers to delivery systems that release the drug only after a lag phase programmed to cover relatively consistent small intestinal transit time. The lag phase would start after gastric emptying thus allowing colon targeting based on the time-dependent approach. In this sense, Mezavant are quite different dosage forms that be defined as pH-responsive and prolonged-release, respectively.

The authors appreciate your comment. Corrections were made in the text.

Line 62. Please check “reach”

The authors appreciate your comment. The word has been modified in the sentence.

  • 5. Anti-inflammatory drug delivery does not seem to be covered in this section and is not even mentioned in the manuscript title. Thus, it could be left out from the heading and the introductory paragraph of the delivery section.

The authors appreciate your comment. All previously mentions of anti-inflammatories have been removed, including in the heading section.

Line 631-32. Passage “Therefore, the use of the drug in its entirety is not allowed, which leads to opting for a complementary alternative, the coating process” is not fully clear and needs to be rephrased.

The authors appreciate your comment. The sentence has been rewritten.

Line 651. “viability” would not apply to the prebiotic.

The authors appreciate your comment. The word has been modified in the sentence.

Reviewer 2 Report

Comments and Suggestions for Authors

The presented version of the article has been significantly improved. However, the authors did not add information from the publications of 2022-2023. I do not insist on additional changes, if the authors do not consider it necessary to “rejuvenate” the Review, this is their choice.

Comments on the Quality of English Language

I recommend showing the text to a native speaker for polishing.

Author Response

The authors welcome your comments during the review process.

Reviewer 4 Report

Comments and Suggestions for Authors

The author made very detailed modifications according to my opinion. Therefore, I recommend that this manuscript be published in its current form.

Author Response

(The authors gave the same response as above.)
